# Cartilage Regeneration Using Human Umbilical Cord Blood Derived Mesenchymal Stem Cells: A Systematic Review and Meta-Analysis

**DOI:** 10.3390/medicina58121801

**Published:** 2022-12-06

**Authors:** Dong Hwan Lee, Seon Ae Kim, Jun-Seob Song, Asode Ananthram Shetty, Bo-Hyoung Kim, Seok Jung Kim

**Affiliations:** 1Department of Orthopedic Surgery, Yeouido St. Mary’s Hospital, College of Medicine, The Catholic University of Korea, 10, 63-ro, Seoul 07345, Republic of Korea; 2Department of Orthopaedic Surgery, Uijeongbu St. Mary’s Hospital, College of Medicine, The Catholic University of Korea, 271, Cheonbo-Ro, Uijeongbu-si 11765, Republic of Korea; 3Department of Orthopaedic Surgery, Gangnam JS Hospital, Seoul 06259, Republic of Korea; 4Institute of Medical Sciences, Faculty of Health and Wellbeing, Chatham Maritime, Canterbury Christ Church University, Kent ME4 4UF, UK

**Keywords:** human umbilical cord blood derived mesenchymal stem cells, cartilage regeneration, cartilage repair, osteoarthritis treatment, stem cell therapy

## Abstract

*Background and Objectives:* Human umbilical-cord-blood-derived mesenchymal stem cells (hUCB-MSCs) have recently been used in clinical cartilage regeneration procedures with the expectation of improved regeneration capacity. However, the number of studies using hUCB-MSCs is still insufficient, and long-term follow-up results after use are insufficient, indicating the need for additional data and research. We have attempted to prove the efficacy and safety of hUCB-MSC treatment in a comprehensive analysis by including all subjects with knee articular cartilage defect or osteoarthritis who have undergone cartilage repair surgery using hUCB-MSCs. We conducted a meta-analysis and demonstrated efficacy and safety based on a systematic review. *Materials and Methods:* This systematic review was conducted following the Preferred Reporting Items for Systematic Reviews and Meta-Analysis (PRISMA) guidelines. For this study, we searched the PubMed, Embase, Web of Science, Scopus, and Cochrane Library literature databases up to June 2022. A total of seven studies were included, and quality assessment was performed for each included study using the Newcastle–Ottawa Quality Assessment Scale. Statistical analysis was performed on the extracted pooled clinical outcome data, and subgroup analyses were completed. *Results:* A total of 570 patients were included in the analysis. In pooled analysis, the final follow-up International Knee Documentation Committee (IKDC) score showed a significant increase (mean difference (MD), −32.82; 95% confidence interval (CI), −38.32 to −27.32; *p* < 0.00001) with significant heterogeneity (I^2^ = 93%, *p* < 0.00001) compared to the preoperative score. The Western Ontario and McMaster Universities Osteoarthritis Index (WOMAC) scores at final follow-up were significantly decreased (MD, 30.73; 95% CI, 24.10–37.36; *p* < 0.00001) compared to the preoperative scores, with significant heterogeneity (I^2^ = 95%, *p* < 0.00001). The visual analog scale (VAS) score at final follow-up was significantly decreased (MD, 4.81; 95% CI, 3.17–6.46; *p* < 0.00001) compared to the preoperative score, with significant heterogeneity (I^2^ = 98%, *p* < 0.00001). Two studies evaluated the modified Magnetic Resonance Observation of Cartilage Repair Tissue (M-MOCART) score and confirmed sufficient improvement. In a study analyzing a group treated with bone marrow aspiration concentrate (BMAC), there was no significant difference in clinical outcome or M-MOCART score, and the post-treatment International Cartilage Repair Society (ICRS) grade increased. *Conclusion:* This analysis demonstrated the safety, efficacy, and quality of repaired cartilage following hUCB-MSC therapy. However, there was no clear difference in the comparison with BMAC. In the future, comparative studies with other stem cell therapies or cartilage repair procedures should be published to support the superior effect of hUCB-MSC therapy to improve treatment of cartilage defect or osteoarthritis.

## 1. Introduction

Osteoarthritis is one of the main diseases of the modern aging population, and available methods for its treatment are gradually expanding with the development of arthroplasty, oral medications, and physiotherapy [1]. Recently, with improvements in quality of life, the target and favored treatment methods of osteoarthritis have changed. Patient needs for early diagnosis and treatment of osteoarthritis are gradually increasing, and many studies are being conducted on joint-preserving surgery. With the introduction of magnetic resonance imaging (MRI), it became possible to evaluate the condition of articular cartilage more precisely before surgery [2], and treatments for articular cartilage defects have advanced. However, due to the limited regenerative capacity of cartilage, no treatment method has achieved a dramatic effect. As cartilage regeneration through microfracture surgery is regenerated by fibrocartilage, its limitations are already well known [3,4]. Accordingly, various methods such as autologous matrix-induced chondrogenesis, autologous chondrocyte implantation, and osteochondral autograft transfer system surgery have been tested for hyaline-like cartilage regeneration [5]. However, problems such as donor site morbidity and the need for multiple operations remain, and problems also persist postoperatively in the long term due to limited cartilage regeneration [6]. In addition to surgical treatment, studies using small-molecule drugs to enhance cell regeneration are also active, which are used not only in cartilage but also in various tissues [7,8].

Basic and clinical research efforts on various types of stem cell therapy are now active. Mesenchymal stem cells (MSCs) have a high regeneration capacity and are most suitable for cartilage repair, and many associated studies have been conducted. MSCs can be obtained from various sources, such as bone marrow, adipose tissue, placenta, and umbilical cords. Methods for extracting MSCs from bone marrow or adipose tissue, such as bone marrow aspiration concentrate (BMAC), adipose-tissue-derived MSCs (Ad-MSCs), and adipose-tissue-derived stromal vascular fraction (ADSVF), have been studied, and their efficacy has been demonstrated [9,10]. However, this treatment approach also involves the need for invasive collection of stem cells, and there are issues with the quantity or quality of stem cells that can be obtained [10]. In contrast, there are fewer problems with the human umbilical cord blood derived MSC (hUCB-MSC) collection process, there are fewer ethical concerns that may arise when using embryonic stem cells, and these cells have a superior differentiation capacity compared to adult stem cells. Accordingly, researchers in various medical fields are increasingly focusing in this direction [11,12,13]. The usage for cartilage repair has also been analyzed in many animal studies [14], and it has been approved as a medical product and is used in clinical practice. Cartistem^®^ (Medipost Inc., Sungnam, Gyeonggi-do, South Korea) was approved by the Korea Food and Drug Administration and is being used in practice, and clinical studies on this product are currently underway in the United States [9]. However, Cartistem^®^ has not yet been used in countries other than Korea, and long-term follow-up results after use are insufficient, indicating the need for additional data and research. Therefore, we aimed to support the efficacy of the cartilage regeneration procedure using hUCB-MSCs as a treatment for cartilage defects through this systematic review. The objective of this systematic review was to prove the efficacy and safety of hUCB-MSCs treatment in a comprehensive analysis by including all subjects with knee articular cartilage defect or osteoarthritis who have undergone cartilage repair surgery using hUCB-MSCs. We included all cartilage repair surgeries without dividing them into arthroscopic assist and open procedure and excluded subjects with injection and other treatments. A meta-analysis synthesizing the clinical outcomes of studies published thus far and an analysis of the collective results were conducted. In some of the included studies, BMAC and microfracture procedures were used as comparison targets for the treatment of cartilage defects, and the results were analyzed. It is expected that the reliability of hUCB-MSC therapy will increase, and more studies will be conducted to achieve more effective treatment methods.

## 2. Materials and Methods

This systematic review was conducted following the Preferred Reporting Items for Systematic Reviews and Meta-Analysis (PRISMA) guidelines [15].

### 2.1. Search Strategy

Two reviewers independently conducted an electronic literature search on the treatment of knee cartilage defects and osteoarthritis using umbilical-cord-blood-derived stem cells. The PubMed, Embase, Web of Science, Scopus, and Cochrane Library databases were searched by two reviewers up to June 2022. The main search keywords were (MeSH term “Cartilage, Articular” or “knee joint” or “Osteoarthritis, knee”) AND (“Umbilical cord blood-derived mesenchymal stem cell” or the MeSH term “Cord Blood Stem Cell Transplantation”), and the search was conducted including additional keywords related to these terms. After the initial search, duplicates were deleted, and each reviewer verified that there were no missing articles from the electronic search (Appendix A).

### 2.2. Study Selection with Eligibility Criteria

The inclusion and exclusion criteria are shown in Table 1. We searched all studies using hUCB-MSCs as a treatment for knee cartilage defects and osteoarthritis. Due to the small number of studies, randomized controlled trials (RCTs), prospective cohort studies, retrospective cohort studies, and case–control studies were all included. Only cases reporting on direct surgical treatment of cartilage lesions using hUCB-MSCs were included, and those involving intra-articular injection were excluded. Animal studies, phase I/II clinical trials, case reports, technical notes, review articles, and articles without accessible full-text versions were excluded, and when two or more studies were published by a single center, any with overlapping patient groups were excluded.

### 2.3. Data Extraction and Quality Assessment

Two reviewers extracted the following data from the included studies: first author, publication year, inclusion criteria, number of participants, age, body mass index, defect size, follow-up duration, outcome, concomitant intervention, and study design. Each extracted data point was verified by the rest of the reviewers. If additional information was needed from the included studies, the author of the study was contacted, and information was obtained.

Quality assessment of the included studies was conducted independently by two reviewers using the Newcastle–Ottawa scale [16]. The study by Lim et al. [17] was evaluated using the Newcastle–Ottawa scale because of the analysis of the clinical outcome during extended follow-up after their RCT. Three quality parameters of the Newcastle–Ottawa scale were evaluated, selection, comparability, and outcome. When the opinions of the reviewers differed, the final decision was achieved through discussion. The final evaluation was divided by the number of stars, and a score ≥ 7 points was indicative of high quality, that of 5–6 points was indicative of moderate quality, and that of four or fewer points was indicative of low quality.

### 2.4. Data and Statistical Analyses

In the included studies, statistical analysis was performed by extracting information on the preoperative clinical outcome and clinical outcome at final follow-up in all cases where hUCB-MSCs were used for treatment. Meta-analysis was performed on the measurement method used simultaneously in three or more studies among the clinical outcome scoring methods. In the study by Lim et al. [17], extended follow-up was performed up to 5 years after the RCT, but the analysis was conducted using the outcome measured at the 3-year follow-up, in line with the follow-up period of other included studies. In the study by Song et al. [18], analysis was performed by dividing the participants into two groups according to the presence or absence of trochlea lesions. Six studies evaluating participants using the International Knee Documentation Committee (IKDC) score, six studies evaluating participants using the Western Ontario and McMaster Universities Osteoarthritis Index (WOMAC), and five studies evaluating participants using the visual analog scale (VAS) were assessed with each measurement method. Each outcome was a continuous variable and was measured as the mean difference (MD) with 95% confidence interval (CI). Heterogeneity was assessed using the I^2^ test. When I^2^ < 50%, a fixed-effects model was used; in other cases, a random-effects model was used. All statistical analyses were performed using RevMan version 5.4 (The Cochrane Collaboration, London, UK).

## 3. Results

### 3.1. Identification of Studies

Figure 1 presents the search information and shows a PRISMA flow diagram of the study selection process. An electronic literature search found 202 articles, and 53 duplicate articles were removed. After checking the titles/abstracts of the remaining 149 articles, those that met the exclusion criteria were removed, and the full-text versions of the remaining articles (n = 14) were analyzed. Among them, seven articles were additionally removed based on the inclusion/exclusion criteria, and seven articles were finally reviewed and meta-analyzed.

### 3.2. Characteristics of Included Studies

Table 2 summarizes the characteristics of the seven studies included in this review, including inclusion criteria, number of participants, age, body mass index, defect size, follow-up duration, outcome, concomitant intervention (performance of high tibial osteotomy (HTO)), and study design. In four studies, HTO was performed concomitantly [18,19,20,21], HTO was not performed in two studies [17,19], and one study did not consider HTO [20]. In two studies, only kissing lesions in the medial compartment were used as inclusion criteria [21,22]. In two studies, the results were evaluated using a modified Magnetic Resonance Observation of Cartilage Repair Tissue (M-MOCART) score generated with MRI [19,20]. All seven studies included in the analysis scored ≥ 6 points when evaluated using the Newcastle–Ottawa scale. Four studies scored 6 points (moderate quality), two studies scored 7 points, and one study scored 8 points (high quality) (Table 3).

### 3.3. Subgroup Analysis of Included Studies

Subgroup analysis was performed in five studies, which are described in Table 2. Analysis of these studies showed that age had no effect on outcome [17,18,19,20], except in one study [21]. In addition, subgroup analysis of lesion size revealed that it did not affect the outcome in three studies [17,18,19] but did so in one study [21]. Unlike in other studies, subgroup analysis revealed that a younger age and a larger lesion size led to significantly greater improvements in the 2-year outcome in the study by Song et al. [21]. However, as the inclusion criteria of this study included age >60 years and kissing lesions in the medial compartment, the age cutoff for dividing the younger/older age groups was 65 years. As the regenerative ability of those > 65 years of age is significantly reduced, this result should be considered separately from the results of the other four studies. In addition, the number of patients included in this study was rather small (n = 25 participants). Therefore, it would be reasonable to conclude that age and lesion size have little effect on outcome based on the collected research results. In two studies [18,19], subgroup analyses were performed on other factors, such as lesion location and number, and neither the presence of trochlear lesions nor the number of lesions had a significant effect on the outcome. However, medial femoral condyle (MFC) lesions led to a significantly worse outcome than trochlear lesions [19]. In addition, subgroup analysis was performed considering obesity in two studies [18,21], showing that obesity did not have a significant effect on the outcome.

### 3.4. Study Outcome

The outcomes of each included study are summarized in Table 4. Publication bias was evaluated using a funnel plot, which showed a symmetrical appearance, suggesting low possibility of bias (Appendix A).

#### 3.4.1. IKDC Score

The preoperative and final follow-up scores were extracted from six studies evaluating IKDC scores and analyzed. Totals of 441 patients with preoperative scores and 431 patients with final follow-up scores were included. The shortest final follow-up period was a mean of 1.7 years, and most studies presented 2–3 years of follow-up data (Table 2). However, in the study by Lim et al. [17], 3-year follow-up data were extracted and used for analysis. In the study by Song et al. [18], the divided groups were evaluated according to the presence or absence of trochlear lesions. In pooled analysis, the final follow-up IKDC score was significantly increased (MD, −32.82; 95% CI, −38.32 to −27.32; *p* < 0.00001) compared to the preoperative score, with significant heterogeneity (I^2^ = 93%, *p* < 0.00001) (Figure 2).

#### 3.4.2. WOMAC Score

The preoperative and final follow-up WOMAC scores were extracted from six studies and analyzed. For this, 446 patients with preoperative scores and 436 patients with final follow-up scores were included. The shortest final follow-up period was a mean of 15.6 months, and most studies presented 2–3 years of data (Table 2). As before, 3-year follow-up data were used from the study by Lim et al. [17], and the data of Song et al. [18] were divided into two groups and analyzed. In pooled analysis, WOMAC final follow-up scores were significantly decreased (MD, 30.73; 95% CI, 24.10–37.36; *p* < 0.00001) compared to preoperative scores, with significant heterogeneity (I^2^ = 95%, *p* < 0.00001) (Figure 3).

#### 3.4.3. VAS Score

The preoperative and final follow-up VAS scores were extracted from five studies and analyzed. Here, 348 patients with preoperative scores and 338 patients with final follow-up scores were included. The final follow-up period ranged from a minimum of 2 years to a maximum of 36.1 months (Table 2). The data of the studies [17,21] in which VAS scores were evaluated based on a total of 100 points were divided by 10, and the standard was unified into a total of 10 points for analysis. As before, 3-year follow-up data were used from the study by Lim et al. [17], and the data of Song et al. [18] were divided into two groups and analyzed. In pooled analysis, the VAS final follow-up scores were significantly decreased (MD, 4.81; 95% CI, 3.17–6.46; *p* < 0.00001) compared to preoperative scores, with significant heterogeneity (I^2^ = 98%, *p* < 0.00001) (Figure 4).

#### 3.4.4. M-MOCART Score

The M-MOCART score was measured in two of the seven studies included in the analysis (Appendix A). In the study by Song et al. [19], 34 patients received M-MOCART scores, and the mean score at 3–6 months after surgery was 30.58 points. According to MRI scans performed ≥ 1 year after surgery (mean, 21.2 months), the mean score increased to 55.44 points. Ryu et al. [20] measured the M-MOCART score using MRI scans at 1 year and 2 years after surgery in 27 patients, recording mean respective scores of 69.63 and 73.7 points. In both studies, the M-MOCART score after ≥1 year was sufficient compared to the general results of other cartilage repair surgeries [24,25]. However, evidence confirming whether the M-MOCART score correlates with clinical outcome is insufficient, and the M-MOCART score should be considered an auxiliary indicator of repaired cartilage quality [25]. In the study by Ryu et al. [20], the M-MOCART scores of 25 patients in the BMAC group were measured and compared with those of the hUCB-MSC group. Ultimately, the results of the BMAC group at 1 year and 2 years were 65.4 and 70.2 points, with the hUCB-MSC group showing better results, although there was no significant difference.

#### 3.4.5. Comparison with BMAC and Microfracture Procedures

Comparisons with a control group were performed in three studies. In one study, the group only treated with microfracture surgery was set as a control group [17]; in two studies, the group treated with BMAC procedures was set as a control group [20,22]. The clinical outcome data of the control group are included in Table 4. In the study by Lim et al. [17], compared to the microfracture group, IKDC scores at 48 weeks did not show a significant difference in the hUCB-MSC group, with a mean of 53.4 points versus a mean of 53.5 points. However, at 3 years of follow-up, the hUCB-MSC group had a mean score of 57.4 points and the microfracture group had a mean score of 49.0 points, confirming that better results were obtained in the hUCB-MSC group. Moreover, this trend continues in the 4- and 5-year follow-up comparisons. WOMAC scores also showed no significant difference between the groups, with a mean of 24.7 points for the hUCB-MSC group and a mean of 26.2 points for the microfracture group at 48 weeks; in contrast, at 3 years of follow-up, the hUCB-MSC group had a mean score of 25.4 points versus the microfracture group mean score of 34.5 points, indicating better results in the hUCB-MSC group. Again, this trend persists in the 4- and 5-year follow-up comparisons. In addition, considering the VAS scores, the results of the hUCB-MSC group appear better in 3-, 4-, and 5-year follow-up comparisons with the microfracture group. Regardless of the scoring system, a divergence between groups seems to appear from the third year onward; although the microfracture group had good results until 48 weeks, this good result was not maintained thereafter. In the study by Ryu et al. [20], the outcome of hUCB-MSC therapy at 2 years of follow-up was compared with that of BMAC, and the hUCB-MSC group had a mean IKDC score of 81.35 points compared to the 80.287 points of the BMAC group, showing no significant difference. There was also no significant difference in VAS scores, with means of 0.85 and 0.92 points, respectively. In the study by Lee et al. [22], the outcome of the hUCB-MSC group was evaluated at a mean of 15.6 months, while that of the BMAC group was evaluated at a mean of 20.7 months. WOMAC scores were slightly better in the hUCB-MSC group, with a mean of 19.5 points, compared to the BMAC group with a mean of 23.4 points, but there was no significant difference. In addition to the WOMAC score, the Hospital for Special Surgery score and the pain and function Knee Society scores were also calculated but did not show significant differences between groups, although the results of the hUCB-MSC group were slightly better.

## 4. Discussion

In the past, arthroplasty was the most common treatment for osteoarthritis; recently, joint-preserving surgery has been developed. In particular, in Korea, the cartilage regeneration procedure performed together with correction of varus or valgus alignment through corrective osteotomy is gradually increasing in popularity [26]. In line with this, stem cell therapy for cartilage defects and osteoarthritis has developed through many studies. Previously, stem cell therapy mainly used BMAC, AD-MSCs, and ADSVF, but research on hUCB-MSCs has been active since Cartistem^®^ was approved by Korea’s Food and Drug Administration. As there is a large number of MSCs in umbilical cord blood, its use has been expanded in various fields [27]. To date, a disadvantage to using stem cells has been that the procedure is invasive, and the cells are difficult to obtain in sufficient number from bone marrow and adipose tissue. On the other hand, sourcing hUCB-MSCs can result in collection of a constant and sufficient number of stem cells in a non-invasive manner. In addition, these cells are hypoimmunogenic and do not cause immune-related problems. In addition, unlike with embryonic stem cells, there are no ethical problems [5,28]. Finally, hUCB-MSCs boast higher proliferation, karyotype stability after prolonged expansion, and easier chondrogenic differentiation compared to MSCs extracted from bone marrow and adipose tissue [29]. Therefore, cartilage regeneration therapy using hUCB-MSCs is considered more effective than conventional treatment and is being used in clinical trials. Many retrospective studies have been announced and usually report satisfactory results. Several case reports have also been published, one of which reported satisfactory results using hUCB-MSCs as a treatment for juvenile osteochondritis dissecans [30].

Therefore, we summarized the published therapeutic effects of hUCB-MSCs and present our opinions. First, hUCB-MSC therapy is a new clinical treatment, and its safety must be confirmed. There appear to be no severe adverse effects (AEs) among the studies published to date; however, there were reports of general AEs in three studies. In the study by Park et al. [28], there were no serious AEs, and one patient experienced an elevation of antithyroglobulin antibody level, which was reported as a treatment-emergent AE, but it spontaneously normalized without any special treatment. Ryu et al. [20] reported three cases of adhesions, but this is a minor complication of arthroscopic surgery, and the frequency in this study was not particularly high. Chung et al. [23] reported that temporary knee swelling was observed up to 1 month after surgery in some patients, but it was self-limiting, and there were no other serious AEs. In the study by Lim et al. [17], there were no other surgical-related serious AEs other than surgical site pain, the final reported serious AEs were also not related to the use of hUCB-MSCs, and there were no immunological reactions. Therefore, the safety of hUCB-MSCs is supported by the available research results.

In several studies included in this review, subgroup analyses were conducted. In four studies with subgroup analyses according to age [17,18,19,20], excluding one other study [21], age did not affect the results. In three studies [17,18,19], lesion size did not affect the results. Considering the process of obtaining stem cells from bone marrow or adipose tissue, there may only be a small number of stem cells extracted from a patient of old age, and there may be an overall decrease in their quality with a low differentiation potential. However, as mentioned earlier, hUCB-MSC therapy involves using a product of a certain quality and number of stem cells, and the results can be seen above. However, implanted MSCs are known to promote the regeneration of host cells with a paracrine effect rather than turning them into cartilage, so it may not be described only by stem cell quality [31,32]. In addition, some studies have reported good pain level and functional outcome when using high-dose MSCs [33], indicating the need for additional research.

In pooled data analysis of seven studies included in this systematic review, IKDC, WOMAC, and VAS scores all showed statistically significant improvements at final follow-up compared to the preoperative evaluation. This means that clinical outcomes can be effectively improved through cartilage regeneration surgery using hUCB-MSCs. The final follow-up duration of the included studies varied, but the mean is ≥15.6 months, and considering that most of the studies included a follow-up period of about 2 years, this means that the clinical outcome of the hUCB-MSC therapy was sufficiently improved and maintained during the short-term follow-up. It is known that the clinical symptoms recur before two years if there is a failure in cartilage regeneration. Therefore, a favorable short-term outcome is implied when the improvement of clinical outcome is maintained during the 2-year follow-up [34]. In addition, three studies [17,18,19] showed good results in clinical outcomes measured after a mean period of ≥3 years. Several studies reported good quality of regenerated cartilage for up to 2 years after microfracture surgery and then gradually deteriorated [35,36]. In the study by Lim et al. [17], the microfracture group had a worse outcome from the third year onward, while the hUCB-MSC group maintained an improved outcome until 5 years of follow-up. With hUCB-MSC therapy, it can be inferred that the quality of regenerated cartilage is good, and regenerated cartilage is more hyaline-like than microfracture alone. In addition, Suh et al. [37] compared radiological changes after surgery between a group that received hUCB-MSC therapy and a group that received microfracture surgery, with both groups receiving HTO as concomitant surgery, and found that the hUCB-MSC group had a significant increase in joint space width increment (*p* < 0.05) of 0.6 mm compared to that of just 0.1 mm in the microfracture group.

Among the studies included in this systematic review, two [20,22] considered BMAC, and no significant differences were seen in the IKDC, WOMAC, VAS, and M-MOCART scores. However, Lee et al. performed second-look arthroscopy in all patients, and the hUCB-MSC group had better ICRS grade results in the medial femoral condyle and medial tibial condyle, so it seems that hUCB-MSC therapy facilitates better cartilage regeneration. Ryu et al. also performed second-look arthroscopy in some patients and measured the ICRS grade, and they reported no significant difference. However, the study by Ryu et al. includes a difference in the mean age of the hUCB-MSC group and BMAC group (53.93 vs. 39.64 years). In other words, the BMAC group may have experienced better results because younger people have better healing potential. Considering these points, more data must be accumulated to compare the clinical outcomes of cartilage regeneration. In the study by Yang et al. [38], which was excluded from this analysis because there was a risk of overlapping patient groups, the results of hUCB-MSC and BMAC treatments were compared by performing propensity score matching for sex, age, body mass index, and lesion size. No significant differences between the two groups in the measurement of IKDC score, Knee Injury and Osteoarthritis Outcome Score, Short-Form 36 score, or Tegner Activity Score were found. However, following ICRS Cartilage Repair Assessment grading, which was carried out through second-look arthroscopy, the score of the hUCB-MSC group (n = 44) was 9.2 ± 2.2 points, and that of the BMAC group was (n = 37) 7.2 ± 3.0 points. In other words, the hUCB-MSC group experienced significantly better cartilage regeneration. Based on these results, hUCB-MSC therapy can be considered to lead to slightly better-quality cartilage regeneration than BMAC. However, there is still little comparative research with BMAC, and more is needed, with long-term results ≥3 years considered crucial.

Our study has some limitations. First, among the studies included in our analysis, six were retrospective studies, and one was an extended follow-up data analysis after RCT. Meta-analysis based on RCT studies could not be performed due to the small number of studies. In order to form a more scientific basis for the effect of hUCB-MSCs, studies analyzing multiple RCTs will be needed. Second, in the studies included in our investigation, several patient characteristics such as age, sex, defect size, and follow-up duration varied, and it is believed that this may have affected the heterogeneity of the results. Lastly, three of the studies included in this meta-analysis were published by a single center, and two of the remaining studies were also published by another single center. We checked the inclusion criteria with the study authors to confirm that there were no overlapping patients and proceeded with our analysis; however, this is a potential cause of bias. All of these limitations are considered to be caused by the insufficient number of studies published so far.

## 5. Conclusions

Cartilage regeneration surgery using hUCB-MSCs is expected to lead to better results than conventional stem cell therapy or other cartilage repair procedures. Through this review, the safety, efficacy, and quality of repaired cartilage associated with this procedure were demonstrated. In the future, a number of comparative studies considering other stem cell therapies or cartilage repair procedures will be published, and we expect that the relatively superior effect of hUCB-MSC therapy will be demonstrated. Based on this, we hope that the treatment of cartilage defects and osteoarthritis will be improved.

## Figures and Tables

**Figure 1 medicina-58-01801-f001:**
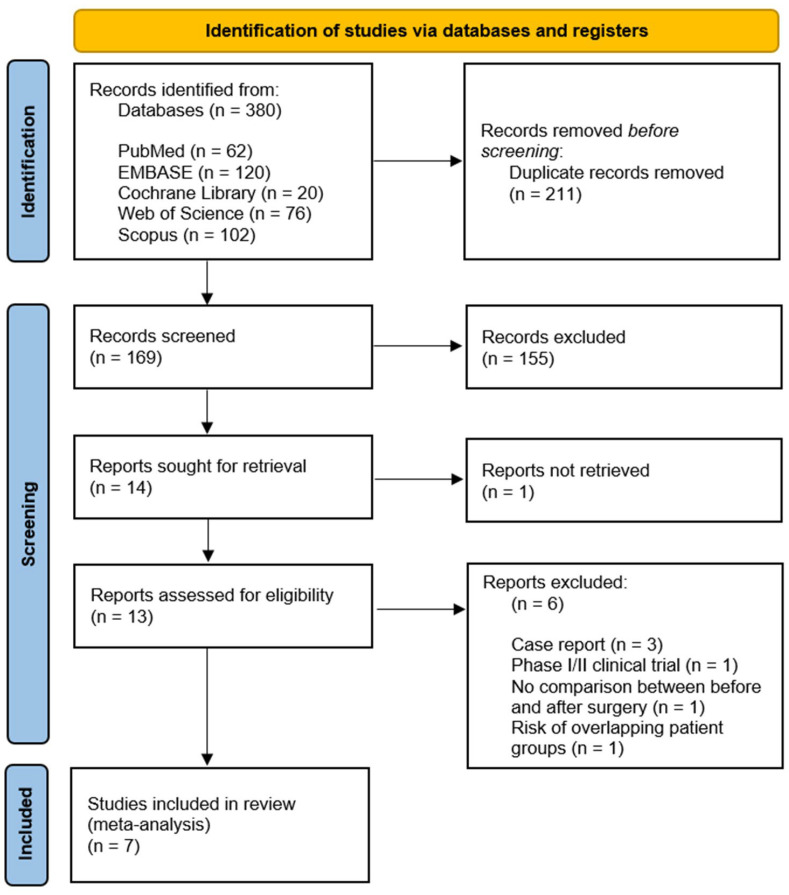
PRISMA flow diagram for the systematic review.

**Figure 2 medicina-58-01801-f002:**
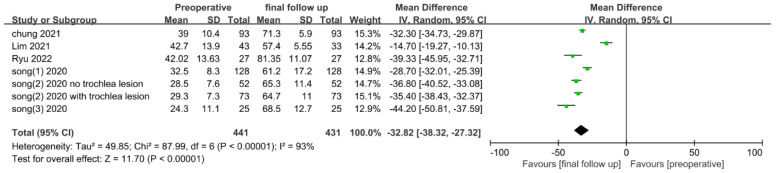
Forest plot of mean differences in IKDC subjective scores between pre-operation and at final follow-up.

**Figure 3 medicina-58-01801-f003:**
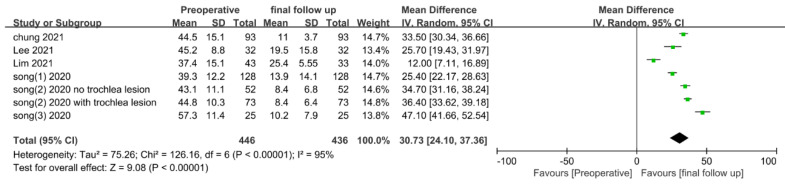
Forest plot of mean differences in WOMAC scores pre-operation and at final follow-up.

**Figure 4 medicina-58-01801-f004:**
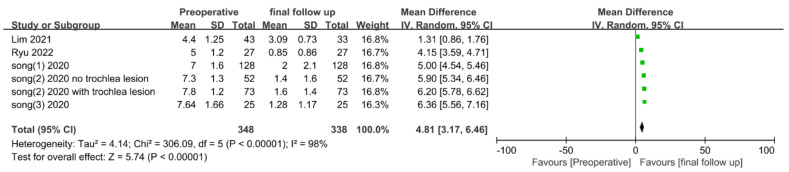
Forest plot of mean difference in VAS scores between pre-operation and final follow-up.

**Table 1 medicina-58-01801-t001:** Inclusion and exclusion criteria.

Inclusion criteria Using hUCB-MSCs as a treatment for knee cartilage defects and osteoarthritis. Only direct surgical treatment of cartilage lesions using hUCB-MSCs. Randomized controlled trials (RCTs), prospective cohort studies, retrospective cohort studies, and case–control studies. Full text available, written in English.
Exclusion criteria Using hUCB-MSCs for intra-articular injection therapy. If there is no clinical outcome comparison between before and after surgery. If there is a risk of overlapping patient groups. Animal studies, phase I/II clinical trials, case reports, technical notes, review articles, and articles without accessible full-text versions.

**Table 2 medicina-58-01801-t002:** Characteristics of included studies.

Study	Inclusion Criteria	Number of Patients		Age, Mean ± SD	BMI, kg/m^2^, Mean ± SD	Defect, cm^2^, Mean ± SD	Follow-Up	Outcome	Concomitant Intervention (HTO)	Study Design	etc.
		Total	Each Group								
chung 2021 [23]	younger than 65 years, ICRS grade III or IV cartilage defects (>2 cm^2^), mechanical femorotibial varus angles > 3°, and KL grade 3	93	N/A	56.6 years (43–65)	25.8 kg/m^2^ (20.9–33.2)	median 6.5 cm^2^ (2.0–12.8)	mean 1.7 Y (1.0–3.5 Y)	IKDC, WOMAC, KSS, HSS, and ICRS	with HTO	retrospective cohort	N/A
song 2020 (1) [19]	older than 40 years, ICRS IV (>2 cm^2^), KL grade 1–3, and femorotibial angle (varus or valgus) < 8°	128	N/A	56.5 ± 7.9 (40–78)	24.6 ± 3.6 kg/m^2^ (17–45.8)	one/two/three, 67 (4.5 ± 1.3)/49 (7.3 ± 2.9)/12 (9.8 ± 3.6)	36.1 ± 6.4 M (25–47 M)	VAS, WOMAC, IKDC, and MOCART (for 34 pts)	without HTO	retrospective cohort	subgroup analysis: trochlea lesion, age, and lesion size
song 2020 (2) [18]	older than 40 years, ICRS IV (>4 cm^2^) in medial compartment, KL grade 1–3, and femorotibial angle varus > 5°	125	with trochlea lesion: 73	58.3 ± 6.8 (43–74)	25.6 ± 2.7 kg/m^2^ (19.2–35.5)	6.9 ± 2 cm^2^	3 Y	VAS, WOMAC, IKDC, and ICRS	with HTO	retrospective cohort	subgroup analysis: age, obesity, lesion size, location, and number of lesions
			without trochlea lesion: 52								
song 2020 (3) [21]	older than 60 years with a kissing lesion of the medial compartment, full-thickness chondral defect ≥ 4 cm^2^ of MFC, and varus deformity ≥3°	25	N/A	64.9 ± 4.4 (60–76)	24.9 ± 3.1 kg/m^2^ (19.2–34.2)	total: 9.4 ± 3.1 cm^2^ (5.3–18.9 cm^2^), MFC: 7.2 ± 1.9 cm^2^ (4.2–12.8 cm^2^), and MTC: 2.2 ± 1.1 cm^2^ (0.2–6.1 cm^2^)	26.7 ± 1.8 M (24–31 M)	VAS, WOMAC, IKDC, and ICRS	with HTO	retrospective cohort	with kissing lesion subgroup analysis: age, BMI, and lesion size
Lee 2021 [22]	ICRS ≥ 3B with kissing lesion in medial compartment	74	BMAC: 42	60.7 ± 4.1	26.1 ± 2.8 kg/m^2^	6.5 ± 2.9 cm^2^	20.7 ± 6.1 M	HSS, WOMAC, KSS, and ICRS	with HTO	retrospective cohort	with kissing lesion
			hUCB-MSC: 32	58.1 ± 3.6	26.6 ± 3.0 kg/m^2^	7.0 ± 1.9 cm^2^	15.6 ± 2.8 M				
Ryu 2022 [20]	KL grade ≤ 2, ICRS IV, older than 15 years, and lesion size 2–10 cm^2^ (BMAC 15–50 yrs)	52	BMAC: 25	39.64 ± 9.83	26.19 ± 3.74 kg/m^2^	4.33 ± 1.66 cm^2^		VAS, IKDC, KOOS, and MOCART	5 pts with HTO	retrospective cohort	subgroup analysis based on age (45 yrs)
			hUCB-MSC: 27	53.93 ± 8.6	26.38 ± 3.54 kg/m^2^	4.77 ± 1.81 cm^2^	2 Y		8 pts with HTO		
Lim 2021 [17]	aged > 18 years, full-thickness chondral defect 2–9 cm^2^, ICRS 4, and KL grade 1–3	89	hUCB-MSC: 43	55.3 ± 8.9	25.7 ± 2.8 kg/m^2^	4.9 ± 2.0 cm^2^	2 Y	VAS, IKDC, WOMAC, and ICRS	without HTO	extended study after RCT	subgroup analysis: age, lesion size
			microfracture: 46	54.4 ± 10.8	26.7 ± 3.9 kg/m^2^	4.0 ± 1.8 cm^2^	2 Y				
	numbers of patients (extended follow-up data after RCT)	73	hUCB-MSC: 36	36 M: 33 (3 loss)	48 M: 28 (4 loss, 3 withdrew consent, and 1 AE)	60 M: 29 (3 loss, 3 withdrew consent, and 1 AE)	5 Y				
			microfracture: 37	36 M: 36 (1 loss)	48 M: 30 (6 withdrew consent, 1 reintervention)	60 M: 28 (7 withdrew consent, 2 reintervention)	5 Y				

BMI = body mass index, SD = standard deviation, HTO = high tibial osteotomy, ICRS = International Cartilage Repair Society, KL = Kellgren Lawrence, MFC = medial femoral condyle, RCT = randomized controlled trial, Y = year, M = month, hUCB-MSC = human umbilical cord blood derived mesenchymal stem cell, BMAC = bone marrow aspiration concentrate, IKDC = International Knee Documentation Committee score, WOMAC = Western Ontario and McMaster Universities Osteoarthritis Index, VAS = visual analogue scale, KSS = Knee Society score, HSS = Hospital for Special Surgery, KOOS = Knee Osteoarthritis Outcome score, MOCART = Magnetic Resonance Observation of Cartilage Repair Tissue.

**Table 3 medicina-58-01801-t003:** Newcastle–Ottawa Scale Quality Assessment of included studies.

	Selection				Comparability		Outcome				
Study	Representativeness of the Exposed Cohort	Selection of Non-exposed Cohort	Ascertainment of Exposure	Outcome of Interest	Cohorts	Control for Additional Factor	Assessment of Outcome	Sufficient Follow-Up	Adequacy of Follow-Up	Total	Quality
chung 2021 [23]	*	0	*	*	*	0	*	*	0	6	moderate
song 2020 (1) [19]	*	0	*	*	*	0	*	*	0	6	moderate
song 2020 (2) [18]	*	0	*	*	*	0	*	*	0	6	moderate
song 2020 (3) [21]	*	0	*	*	*	0	*	*	0	6	moderate
Lee 2021 [22]	*	*	*	*	*	0	*	*	0	7	high
Ryu 2022 [20]	*	0	*	*	*	0	*	*	*	7	high
Lim 2021 [17]	*	*	*	*	*	0	*	*	*	8	high

* means the subject gets points in that area

**Table 4 medicina-58-01801-t004:** Clinical outcomes of included studies.

Study	Follow-Up	Treatment and Subgroup	IKDC					WOMAC					VAS				
chung 2021 [23]	mean 1.7 Y (1.0–3.5)	hUCB-MSC	pre	final				pre	final				N/A				
			39.0 ± 10.4	71.3 ± 5.9				44.5 ± 15.1	11.0 ± 3.7								
song (1) 2020 [19]	36.1 ± 6.4 M (25–47)	hUCB-MSC	pre	1 Y	final			pre	1 Y	final			pre	1 Y	final		
			32.5 ± 8.3	55.8 ± 14.3	61.2 ± 17.2			39.3 ± 12.2	17.2 ± 12.7	13.9 ± 14.1			7.0 ± 1.6	2.5 ± 1.7	2.0 ± 2.1		
song (2) 2020 [18]	3 Y	hUCB-MSC	pre	1 Y	2 Y	3 Y		pre	1 Y	2 Y	3 Y		pre	1 Y	2 Y	3 Y	
		trochlear lesion	29.3 ± 7.3	56.7 ± 9.7	61.6 ± 10.7	64.7 ± 11		44.8 ± 10.3	13.8 ± 7.5	10.7 ± 6.9	8.4 ± 6.4		7.8 ± 1.2	2.6 ± 1.7	2.1 ± 1.5	1.6 ± 1.4	
		no trochlear lesion	28.5 ± 7.6	56.7 ± 9.7	61.4 ± 9.5	65.3 ± 11.4		43.1 ± 11.1	13.2 ± 8.2	11.3 ± 8.5	8.4 ± 6.8		7.3 ± 1.3	2.3 ± 1.6	2.0 ± 1.7	1.4 ± 1.6	
song (3) 2020 [21]	26.7 ± 1.8 (24–31)	hUCB-MSC	pre	1 Y	2 Y			pre	1 Y	2 Y			pre	1 Y	2 Y		
			24.3 ± 11.1	58.9 ± 10.3	68.5 ± 12.7			57.3 ± 11.4	15.6 ± 9.6	10.2 ± 7.9			76.4 ± 16.6	20.4 ± 15.1	12.8 ± 11.7		
Lee 2021 [22]								pre	final								
	20.7 ± 6.1 M	BMAC	N/A					43.9 ± 12.7	23.4 ± 11.6				N/A				
	15.6 ± 2.8 M	hUCB-MSC	N/A					45.2 ± 8.8	19.5 ± 15.8				N/A				
Ryu 2022 [20]			pre	final									pre	final			
	2 Y	BMAC	44.17 ± 12.5	80.27 ± 9.48				N/A					5.2 ± 1.1	0.92 ± 0.98			
		hUCB-MSC	42.02 ± 13.63	81.35 ± 11.07				N/A					5.0 ± 1.2	0.85 ± 0.86			
Study	Follow-up	Treatment and Subgroup	IKDC					WOMAC					VAS				
Lim 2021 [17]			pre	48 weeks	3 Y	4 Y	5 Y	pre	48 weeks	3 Y	4 Y	5 Y	pre	48 weeks	3 Y	4 Y	5 Y
	5 Y	microfracture	41.8 ± 13.4	53.5 (48.5 to 58.5)	49.0 (43.3 to 54.7)	48.9 (42.1 to 55.7)	47.1 (41.1 to 53.2)	40.4 ± 14.8	26.2 (21.1 to 31.2)	34.5 (27.2 to 41.8)	35.8 (27.6 to 44.1)	36.2 (28.6 to 43.8)	44.6 ± 12.9	24.1 (18.3 to 29.9)	41.1 (32.2 to 50.0)	43.3 (34.7 to 51.8)	43.5 (35.3 to 51.6)
		hUCB-MSC	42.7 ± 13.9	53.4 (49.0 to 57.8)	57.4 (50.8 to 64.1)	53.7 (48.2 to 59.3)	54.7 (48.7 to 60.7)	37.4 ± 15.1	24.7 (20.5 to 28.9)	25.4 (19.9 to 31.0)	28.6 (22.4 to 34.9)	26.9 (20.4 to 33.5)	44.0 ± 12.5	24.2 (17.5 to 31.0)	30.9 (23.6 to 38.2)	35.7 (29.2 to 42.3)	29.1 (22.4 to 35.8)

Y = year, M = month, hUCB-MSC = human umbilical cord blood derived mesenchymal stem cell, BMAC = bone marrow aspiration concentrate, IKDC = International Knee Documentation Committee score, WOMAC = Western Ontario and McMaster Universities Osteoarthritis Index, VAS = visual analogue scale, pre = preoperative, final = final follow-up.

## Data Availability

The data presented in this study are available in the main article and Appendix A.

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
