# Peer review of "Cartilage Regeneration Using Human Umbilical Cord Blood Derived Mesenchymal Stem Cells: A Systematic Review and Meta-Analysis"

_medicina, 2022, doi:10.3390/medicina58121801_

Round 1

Reviewer 1 Report

The systematic review entitled as "Cartilage regenaration using human umbilical cord blood-derived mesenchymal stem cells: A systematic review and meta-analysis" by Dong Hwan Lee et al. is rather interesting seeking an answer to a quite important disease such as osteoarthritis. Indeed, human umbilical cord blood–derived mesenchymal stem cells (hUCB-MSCs) have recently been used in clinical  cartilage regeneration procedures with the expectation of improved regeneration  capacity. Therefore,  a meta-analysis like this can demonstrate the efficacy and the safety of the aforementioned procedures. The research design is appropriate and the methods adequately described. However, I do believe that in the introduction should be added a study based on the in vitro wound healing properties of human cartilage acidic protein demonstrating its ability to regenerate (doi: 10.1016/j.biochi.2020.02.008).

I do believe that this systematic review merits a publication at medicina 

Reviewer 2 Report

The manuscript presents a review related to a relevant topic, i.e., the application of human umbilical cord blood-derived MSCs for cartilage regeneration. The review included in the assessment n.7 studies recently published but the search strategy is limited to a few databases.
1.    Search strategy: As before mentioned, the search strategy should include additional databases such as Web of Science and Scopus to ensure comprehensive identification of studies. The rationale behind the search strategy in terms of the selection of databases is not clear. Moreover, the full search strategies for all databases should be more detailed including any filters, limits, and all keywords used and fields - (ref. lines 124-125).
2.    Methods: PRISMA guidelines (line 115) used is the previous version – 2009, as also indicated in the references. The PRISMA 2020 statement replaces the 2009 statement and includes new reporting guidance (including flow diagram v.2020, etc.). Thus, the new version should be used.
3.    An explicit statement of the objective the review addresses should be provided.
4.    Lines 88-92: I suggest clarifying the text in the mentioned lines, “fewer” ethical issues could be associated with hUCB-MSC than “no problems/concerns” at all.
5.    Line 399: Information about the duration of the final follow-up is relevant and well-discussed. But, the sentence […the effect is sufficient in the short term] should be clarified and it could be interesting to detail any available information from the included studies about a possible quantification of the minimum time resulted as necessary to observe the cartilage regeneration under the considered specific treatment. 

Round 2

Reviewer 2 Report

The manuscript has mostly been improved according to the previous comments. Please, find here below a clarification about my previous comment indicated as Point3 and other aspects to be considered.

Point3: I was referring to the section “Introduction” and to the relevance of adding a specific and detailed statement about the objectives and questions of the review which have only been briefly mentioned (ref. lines 104-106). Thus, an explicit and detailed statement should be included. The PRISMA checklist itself details as an item, “provide an explicit statement of the objective(s) or question(s) the review addresses” (ref. to PRISMA 2020 Checklist). For Table 1, I consider it useful to keep.

Figure1 - PRISMA 2020 flow diagram: I see that the content has been updated however, the original format of the flow diagram 2020 (the template is available online) should be adopted as well; for example, the name of each step of the flow is missing (identification, screening, etc.).

Abstract: The text should be updated including the additional database considered, ref. lines (23-24), and background and objectives should be stated separately and detailed also according to the indications included in the previous Point3.
